# How has the COVID-19 pandemic affected the utilisation of online consultation and face-to-face medical treatment? An interrupted time-series study in Beijing, China

Shan Zhang  , Chengyu Ma

School of Public Health, Capital Medical University, Beijing, China

**Correspondence to**
Chengyu Ma;
101324151@qq.com

## ABSTRACT

**Objective** The COVID-19 pandemic has had a major impact on healthcare utilisation. This study aimed to quantify how the online and face-to-face utilisation of healthcare services changed during this time and thus gain insights into the planning of future healthcare resources during the outbreak of infectious diseases.

**Design** This work is an interrupted time-series study.

**Setting** Monthly hospital-grade healthcare-service data from 22 tertiary first-class public hospitals managed by the Beijing Hospital Authority and online-consultation data from GoodDoctor were used in this study.

**Methods** This is an interrupted time-series study about the change in face-to-face and online healthcare utilisation before and after the COVID-19 outbreak. We compared the impact of COVID-19 on the primary outcomes of both face-to-face healthcare utilisation (outpatient and emergency visits, discharge volume) and online healthcare utilisation (online consultation volume). And we also analysed the impact of COVID-19 on the healthcare utilisation of different types of diseases.

**Results** The monthly average outpatient visits and discharges decreased by 36.33% and 35.75%, respectively, compared with those in 2019 in 22 public hospitals in Beijing. Moreover, the monthly average online consultations increased by 90.06%. A highly significant reduction occurred in the mean outpatients and inpatients, which dropped by 1 755 930 cases (p<0.01) and 5 920 000 cases (p<0.01), respectively. Online consultations rose by 3650 cases (p<0.05). We identified an immediate and significant drop in healthcare services for four major diseases, that is, acute myocardial infarction (−174, p<0.1), lung cancer (−2502, p<0.01), disk disease (−3756, p<0.01) and Parkinson's disease (−205, p<0.01). Otherwise, online consultations for disk disease (63, p<0.01) and Parkinson's disease (25, p<0.05) significantly increased. More than 1300 unique physicians provided online-consultation services per month in 2020, which was 35.3% higher than in 2019.

**Conclusions** Obvious complementary trends in online and face-to-face healthcare services existed during the COVID-19 pandemic. Different changes in healthcare utilisation were shown for different diseases. Non-critically ill patients chose online consultation immediately after the COVID-19 lockdown, but critically ill patients chose

## STRENGTHS AND LIMITATIONS OF THIS STUDY

⇒ Based on the Beijing Hospitals Authority and the GoodDoctor data, this work is an interrupted time-series study to quantify how the online and face-to-face utilisation of healthcare services changed during the COVID-19 pandemic.
⇒ The methodology in exploring the characteristics and trends of changes in healthcare utilisation in the short and long terms during the pandemic is noteworthy.
⇒ Interrupted time series methodologies such as this have limitations in external environmental factors. Other policies and seasonal confounding factors such as meteorological factors may affect medical visits.

hospital healthcare services first. Additionally, the volume of online physician services significantly rose as a result of COVID-19.

## INTRODUCTION

In early 2020, the COVID-19 pandemic affected most of the planet and rapidly changed the utilisation of healthcare services worldwide.[1] Fear of infection and reduced availability of healthcare services has led to decreased non-COVID healthcare utilisation in hospitals.[2 3] The rapid and innovative deployment of telemedicine and online health services has occurred in many countries to relieve the pressure on hospitals and to prevent cross-infection.[4 5] In the USA, emergency room visits dropped by 42% in April, followed by a 26% reduction at the end of May, compared with 2019.[6] Meanwhile, telemedicine visits increased from 102.4 daily to 801.6 daily between 2 March and 14 April 2020.[7]

The impact of the global pandemic on healthcare systems is acute. As the COVID-19 pandemic continues, many studies have

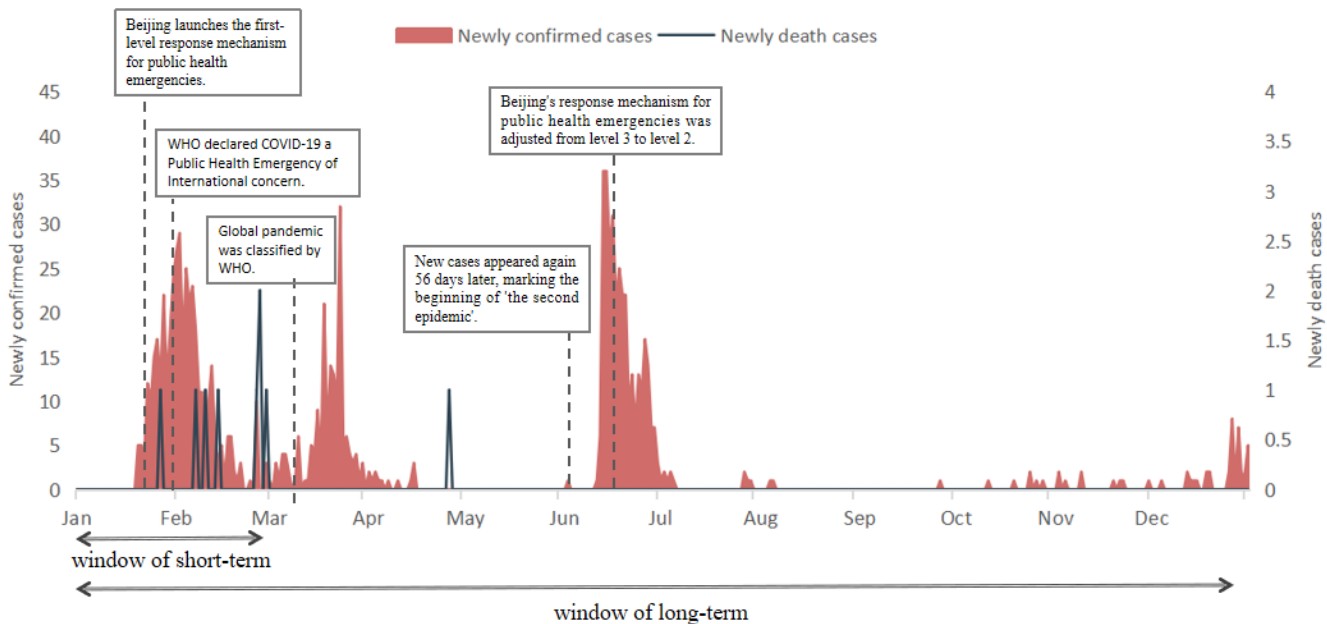

**Figure 1** COVID-19 epidemic trend in Beijing.

reported major changes in the utilisation of healthcare services because of measures such as lockdowns and stay-at-home orders. These changes include large reductions in hospital visits, and some selective increases in online health services.[8–10] However, minimal data are available on the relationship between hospital visits and online health services over the same period, especially with quantitative time-trend evidence. Examining the short-term and long-term patterns of changes in hospital volume and online health services during the COVID-19 pandemic is valuable. The extent and effects of substitution with online healthcare services for different kinds of diseases also require further investigation.

Beijing is a megacity with the most extensive internet accessibility (77.8%)[11] and the most famous hospitals compared with other parts of China. After the COVID-19 pandemic, many patients were unable to access hospitals in Beijing owing to the lockdown policies. In response, online health services and an internet-based health system have been utilised by hospitals in addition to third-party service platforms, such as GoodDoctor (www.haodf.com). Physicians are active in spending their free time on these platforms to provide patients with high-quality online health consultations. Online consultations refer to patients using the Internet to communicate with physicians in real-time through online text, pictures, video, audio, etc.

Figure 1 depicts the epidemic trends and significant changes in the COVID-19 outbreak in Beijing. The first confirmed case of COVID-19 in Beijing occurred on 20 January 2020. Since then, the virus has spread across Beijing. Beijing launched the first-level response mechanism for public health emergencies on 24 January 2020.

To respond to public health emergencies promptly, China has launched a tiered response mechanism based on different pandemic levels and hazards. The first-level response mechanism is unified deployment by provincial headquarters according to the decisions of The State Council. The second-level response mechanism refers to deployment by the provincial headquarters. The third-level response mechanism refers to deployment by the headquarters of the prefect-level city. Two peaks occurred on 2 February and 23 March in the early days. Afterwards, the epidemic entered a normalisation stage, with no new cases for a long time. On 11 June, a 52-year-old man from Xinfadi Market in Beijing was diagnosed with COVID-19. After a 56-day zero-new-case interval, marking the beginning of 'the second epidemic', new cases increased sharply on 13 June. The number of new cases in a single day reached the highest value of 36 cases. Beijing's response mechanism for public-health emergencies was adjusted from level 3 to level 2 on 16 June. This phenomenon led to a fast downward slope with the control of the Chinese government. The epidemic in Beijing has gradually stabilised since then. As of 31 December, 987 confirmed cases and nine deaths had been reported. In the present study, the change in short term refers to the instantaneous change in medical-service utilisation at the beginning of the pandemic from the end of December 2019 to February 2020, whereas long-term analysis refers to a snapshot change within the first year of the pandemic.

This study was designed to compare and analyse the changing trend of online consultation and face-to-face healthcare volume, as well as the change in different disease types during the pandemic. The research illustrates the speed, scale and reception of

online-consultation utilisation by patients and physicians, as well as the complementary trends in online and face-to-face health services. The results may help provide insights into the planning of future healthcare resources during the outbreak of infectious diseases.

## METHODS
### Study design
We conducted an interrupted time-series analysis to estimate changes in face-to-face and online healthcare utilisation before and after the COVID-19 outbreak. Simple pre–post designs perform before–after comparisons by estimating the change from a single preintervention time point to a single postintervention time point. However, these have poor internal validity as they cannot exclude underlying trends as a cause for any change.[12] Conversely, interrupted time series (ITS) use multiple observations in the preintervention and postintervention periods, thus offer thereby offering a quasi-experimental research design with a potentially high degree of internal validity.[13 14] We treated COVID-19 as the intervention, and 2019 and 2020 as the preintervention and postintervention periods respectively, to analyse the change in face-to-face and online healthcare utilisation after the COVID-19 outbreak.

### Setting
Monthly hospital-grade healthcare-service data from January 2019 to December 2020 were provided by Beijing Hospitals Authority. Beijing municipal hospitals refer to the public hospitals managed by the Beijing Hospitals Authority of Beijing Municipal Health Commission.

### Participants
A total of 22 hospitals were selected in this study, and they covered all tertiary first-class public hospitals managed by the Beijing Hospital Authority. Thirteen were general hospitals and nine were specialised hospitals (obstetrics and gynaecology, paediatrics, oncology, stomatology, chest, psychiatry, traditional Chinese medicine, etc). We collected outpatient and emergency visits, the number of discharges, online consultation volume from all 22 Beijing municipal hospitals, and patients including adults and children. Discharged patients refer to patients who are transferred from outpatient or emergency departments to inpatient departments, including acute, chronic and severe cases.

### Patient and public involvement
This work was an observational cohort study using publicly available data from the Beijing Hospital Authority and GoodDoctor website. Patients were not involved in our design, conduct, or reporting.

### Data sources
First, outpatient and emergency visits, as well as the number of discharges in 2019 and 2020 from all 22 Beijing municipal hospitals were used as the overall face-to-face healthcare utilisation outcomes. Four diseases including acute myocardial infarction (AMI), lung cancer (LC), disk disease and Parkinson's disease were selected to analyse changes in healthcare-service utilisation for different disease severities. These four diseases belonged to the 20 monitored diseases of the Beijing Municipal Health Commission. We used The International Statistical Classification of Diseases and Related Health Problems, Tenth Revision (ICD-10) to identify the principal diagnoses based on the main diagnosis of the electronic medical-record system of hospitals. Owing to data-sharing restrictions, the monthly number of discharges for these four diseases was collected from July 2019 to December 2020.

Second, online consultation data were obtained from GoodDoctor (www.haodf.com). GoodDoctor is the largest and longest-running online physician–patient communication platform in China.[15] It was founded in 2006. As of October 2021, it has collected information on 860 000 physicians from 9780 regular hospitals in China, among which more than 73% are physicians in tertiary hospitals.[16] The 22 hospitals are all included in the GoodDoctor platform, covering more than 80% of outpatient doctors of these hospitals. A large number of patients turned to this platform for help after the COVID-19 pandemic outbreak, largely representing the trend in the utilisation of online consultations during the pandemic.

The platform provides historical data of online consultations from 2010 to the present. We collected the daily online consultation data of the abovementioned 22 public hospitals in 2019 and 2020 and screened out the online-consultation volume data for AMI, LC, disk disease and Parkinson's disease based on disease information. Figure 2 shows the content of the online consultation provided by this platform. We used Java language to design a web crawler and capture daily online-consultation information from web pages. Specific extracted fields include the date, website link, title, disease description, disease name, expected treatment, the duration of illness, consultation text of the patient and physician, and the name, title, department, patient voting numbers, online-consultation volume and satisfaction of online service physicians. The final sample size collected was 70 086 for 2019 and 144 651 for 2020.

### Statistical methods
We fitted an ordinary least squares model by using either regress or Newey (with lag specified) with monthly service volume as the primary outcome of interest. We produced Newey–West SEs to handle autocorrelation. We treated COVID-19 as the intervention, and 2019 and 2020 as the preintervention and postintervention periods, respectively. The ITSA regression model is expressed as follows:

$$Y_t = \beta_0 + \beta_1 T_t + \beta_2 X_t + \beta_3 X_t T_t + \varepsilon_t$$

where $Y_t$ is the effect variable, including outpatient and emergency visits, discharge volume and online-consultation volume; $T_t$ is the time since the start of the

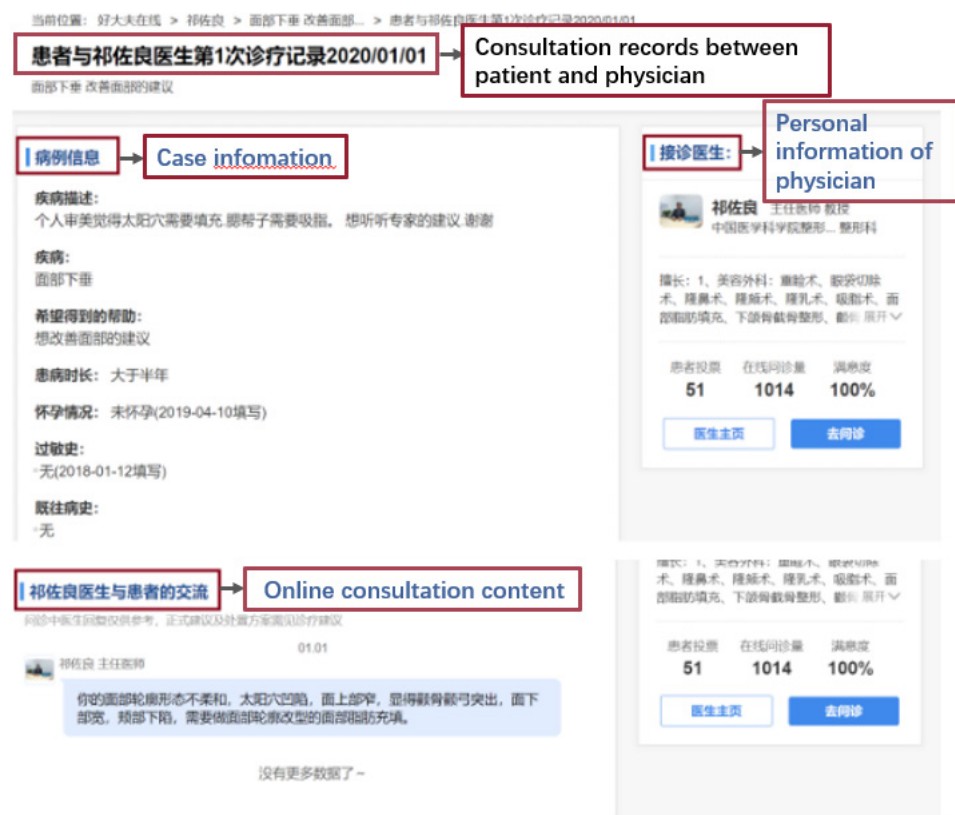

**Figure 2** Information on online consultation provided by this platform.

study, $X_t$ is a dummy variable representing the intervention (preintervention period 0, otherwise 1) and $X_t T_t$ is the interaction term; $\beta_0$ is the intercept or starting level of the outcome variable; β1 is the slope or trajectory of the outcome variable until the COVID-19 outbreak; $\beta_2$ is the change in the level of outcome that occurred in the period immediately following the COVID-19 outbreak (compared with the counterfactual) and $\beta_3$ is the difference between the prepandemic and pandemic slopes of the outcome. Accordingly, we searched for significant p-values in $\beta_2$ to indicate an immediate treatment effect or in $\beta_3$ to indicate a treatment effect over time.

Moreover, we determined the impact of COVID-19 on the utilisation of healthcare services for different diseases.

We stratified models by disease type restricted to (1) AMI, (2) LC, (3) disk disease and (4) Parkinson's disease. They were included in the 20 monitored diseases of Beijing Hospital Authority with clear diagnostic criteria. These four diseases had a certain number of patients online and face-to-face. AMI and LC were considered severe cases, whereas disk disease and Parkinson's disease were considered mild cases in the research. All statistical analyses were conducted using Stata V.15.0 and SPSS V.20.0.

## Bias

Although ITS has high internal validity, external confounding factors cannot be controlled. Since the outbreak of COVID-19 across the entirety of China in 2020, we cannot find an effective control group. A certain confounding bias existed in this study.

## RESULT
### Impact of the COVID-19 pandemic on the utilisation of healthcare services

Table 1 shows the descriptive statistical indicators of the face-to-face outpatient and emergency visits, discharges and online consultations in 2019 and 2020. Compared with 2019, the utilisation of face-to-face healthcare services decreased in 2020. The monthly average outpatient and emergency visits, as well as discharges, decreased by 36.33% and 35.75%, respectively. Compared with 2019, the utilisation of online consultations increased in 2020. The monthly average online consultations increased by 90.06%.

Figure 3 shows the percentage changes in healthcare-service utilisation in Beijing. Compared with the same period in 2019, we observed lower monthly face-to-face outpatient and emergency visits, as well as discharges in 2020. The percentage change frequency for both was the greatest in March, with 66.97% and 75.17% decrease, respectively. The overall change decreases thereafter. Notably, when affected by 'the second epidemic', face-to-face healthcare services increased slightly again, but utilisation in July fell at a higher rate than in June. Online consultations in all months except January were higher

**Table 1** Healthcare-service utilisation in 2020 compared with 2019 (10 000)

|  | 2019 (before the COVID-19 outbreak) | | | 2020 (after the COVID-19 outbreak) | | |
|---|---|---|---|---|---|---|
|  | Face-to-face outpatient and emergency visits | Face-to-face discharges | Online consultation | Face-to-face outpatient and emergency visits | Face-to-face discharges | Online consultation |
| Mean | 270.179 | 8.677 | 0.634 | 172.018 | 5.575 | 1.205 |
| Std | 25.779 | 0.918 | 0.054 | 54.554 | 2.160 | 0.238 |
| Median | 274.391 | 8.956 | 0.611 | 179.315 | 5.996 | 1.272 |
| Max | 311.549 | 9.559 | 0.754 | 243.728 | 8.292 | 1.450 |

than in the same period in 2019. The overall change increased. The greatest decrease was observed in August. Figure 3 shows that in the early days of the pandemic (January to June), the utilisation of online and face-to-face healthcare services showed obvious complementary trends. Afterwards, the growth rate of online consultations remained relatively high, whereas the decline rate of face-to-face healthcare services gradually decreased. This finding suggested that the utilisation of face-to-face healthcare services is gradually returning to normal levels as COVID-19 fades away, whereas online consultations is maintaining a high growth rate.

## Interrupted time-series analysis
### Changes in overall online and face-to-face healthcare services

Table 2 shows a highly significant reduction in the mean number of outpatient and emergency visits, as well as discharges, which dropped by 1 755 930 cases (p<0.01) and 59 200 cases (p<0.01), respectively. Before the COVID-19 outbreak, face-to-face healthcare services (including outpatient and emergency visits and discharges) increased (figure 4). Soon after the COVID-19 outbreak, the slope of this trend line significantly increased, but the trend was not statistically significant. Online consultations also rose by 3650 cases (p<0.05), followed by a significant increase

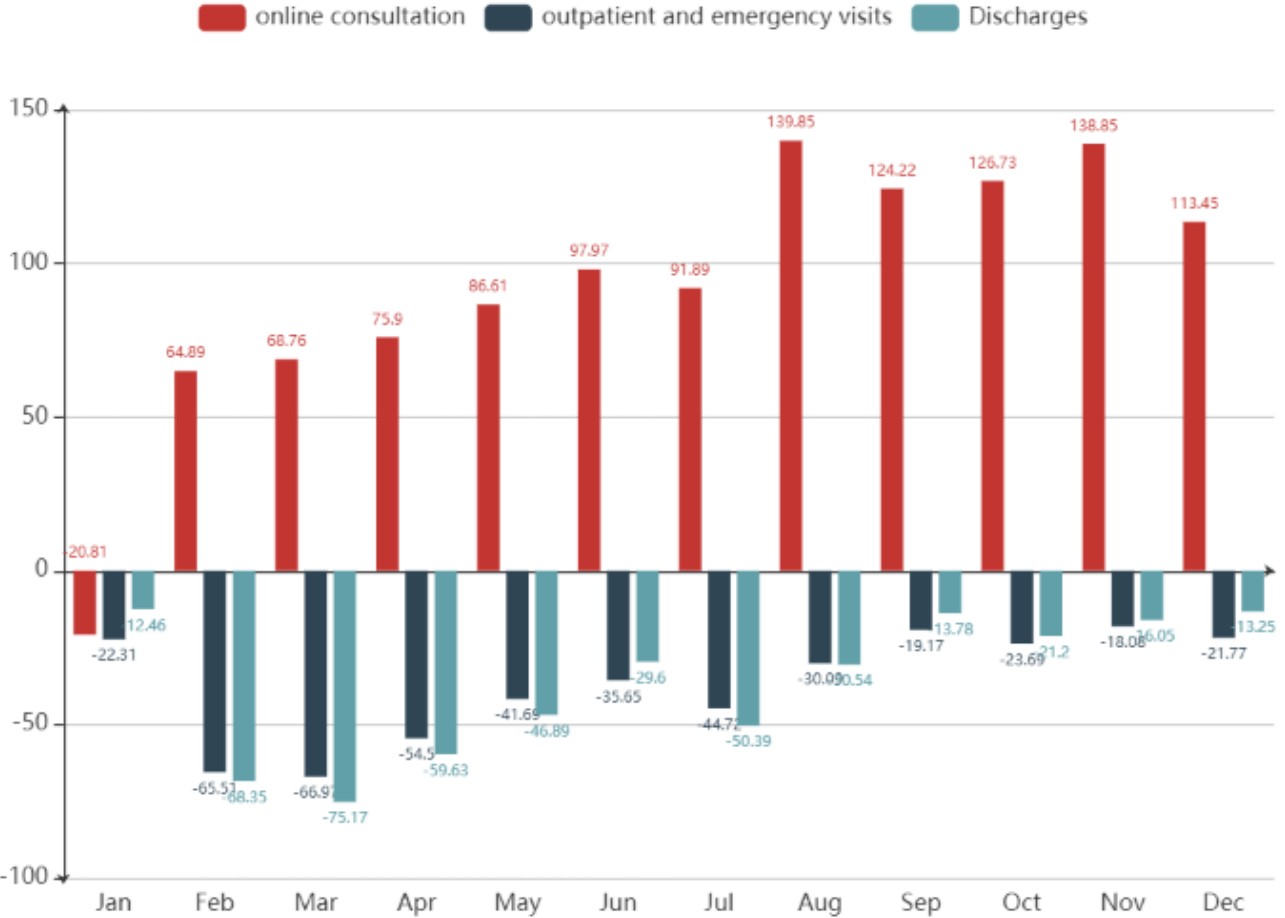

**Figure 3** Percentage change in healthcare-service utilisation in 2020 compared with that in 2019.

**Table 2** Impact of COVID-19 on the utilisation of online and face-to-face healthcare services (10 000)

| | Medical utilisation reduction after the COVID-19 outbreak (intercept) | P value | Average monthly medical utilisation decline after the COVID-19 outbreak (potential trend) | P value |
|---|---|---|---|---|
| Face-to-face outpatient and emergency visits | −175.593 (27.247*) | <0.01 | 7.357 (4.454) | >0.1 |
| Face-to-face discharges | −5.920 (1.608) | <0.01 | 0.283 (0.217) | >0.1 |
| Online consultation | 0.365 (0.143) | <0.05 | 0.058 (0.019) | <0.01 |

* SE in parentheses.

in the monthly trend of utilisation (relative to the pre-pandemic) of 580 cases.

**Changes in face-to-face healthcare services by type of disease**
Table 3 summarises the effect of COVID-19 on face-to-face healthcare services for different diseases. The COVID-19 outbreak was associated with a rapid decline in face-to-face healthcare services for different diseases. From the perspective of immediate effect, we identified a significant drop in healthcare services for the four diseases. From the perspective of long-term effects, AMI, LC and disk disease services showed a nonsignificant stepwise increase (the slope of the trend line increased). Simultaneously, the monthly trend of utilisation for Parkinson's

disease significantly increased (relative to the prepandemic period) (figure 5).

**Changes in online consultations by type of disease**
Table 3 summarises the effect of COVID-19 on online consultations for different diseases. The COVID-19 outbreak was associated with a rapid rise in online consultations for different diseases. From the perspective of immediate effect, we identified a significant increase in healthcare services for disk disease and Parkinson's disease. Conversely, online consultations for AMI and LC showed a small, non-significant decrease. From the perspective of long-term effects, online consultations for AMI, LC and disk disease increased monthly at a rate of

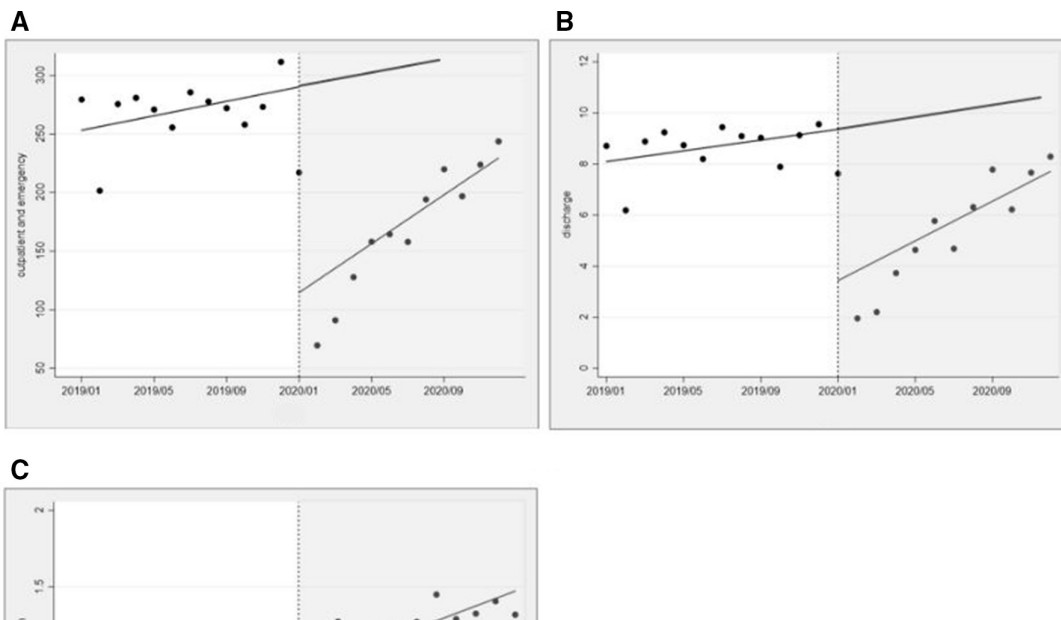

**Figure 4** Effect of COVID-19 intervention on face-to-face and online healthcare utilisation. Scatter plots represent monthly medical utilisation. The black line of best fit represents medical utilisation across estimated by postregression. The vertical black line is the intervention point. (A) Face-to-face outpatient and emergency visits, (B) face-to-face discharges and (C) online consultation.

**Table 3** Impact of COVID-19 on the utilisation of face-to-face healthcare services and online consultation by type of disease

| | Medical utilisation reduction after the COVID-19 outbreak (intercept) | | Average monthly medical utilisation decline after the COVID-19 outbreak (potential trend) | |
| --- | --- | --- | --- | --- |
| | Face-to-face healthcare services | Online consultation | Face-to-face healthcare services | Online consultation |
| Acute myocardial infarction | −174.405 (84.060)* | −5.998 (3.511) | 12.438 (11.540) | 1.633 (0.472)*** |
| Lung cancer | −2502.02 (367.118)*** | −11.832 (19.413) | 80.508 (80.480) | 14.360 (3.079)*** |
| Disk disease | −375.560 (72.369)*** | 63.118 (11.955)*** | 29.163 (16.876) | 3.402 (1.851)* |
| Parkinson's disease | −205.229 (47.852)*** | 24.625 (10.349)** | 12.669 (4.892)** | 1.772 (1.335) |

*P<0.1, **p<0.05, ***p<0.01.
SE in parentheses.

1633, 14 360 and 3402, respectively after the COVID-19 outbreak (figure 6).

### Impact of COVID-19 pandemic on online-service physicians' volume

As affected by COVID-19, the monthly volume of online services provided by physicians in 2020 was significantly higher than that in 2019, except for January. On average, more than 1300 unique physicians from 22 hospitals in Beijing provided online-consultation services per month in 2020, which was 35.3% higher than that in 2019. This finding suggested that the supply of online consultations greatly increased after the COVID-19 outbreak (figure 7).

### DISCUSSION

Based on monthly treatment data from 22 public hospitals in Beijing, obvious complementary trends existed in online and face-to-face healthcare services during the COVID-19 pandemic. We found that public hospitals in Beijing experienced substantial losses in face-to-face healthcare and substantial increases in online-consultation services when the COVID-19 pandemic began in China in 2020. The interrupted time-series models revealed that after the COVID-19 outbreak, the considerable reduction in face-to-face healthcare services was a short-term effect, whereas the increase in online consultation was a

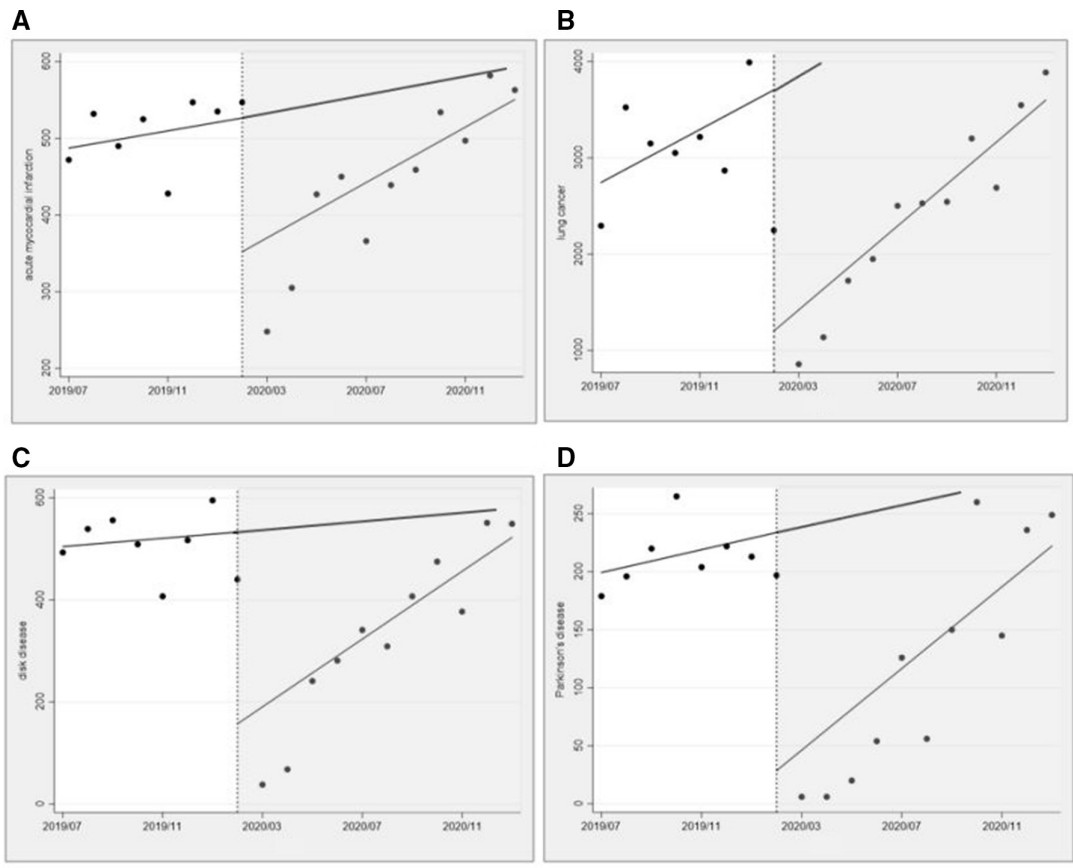

**Figure 5** Effect of COVID-19 intervention on face-to-face healthcare utilisation by type of disease. Scatter plots represent medical utilisation monthly. The black line of best fit represents medical utilisation across estimated by postregression. The vertical black line is the intervention point. (A) AMI, (B) LC, (C) disk disease and (D) Parkinson's disease.

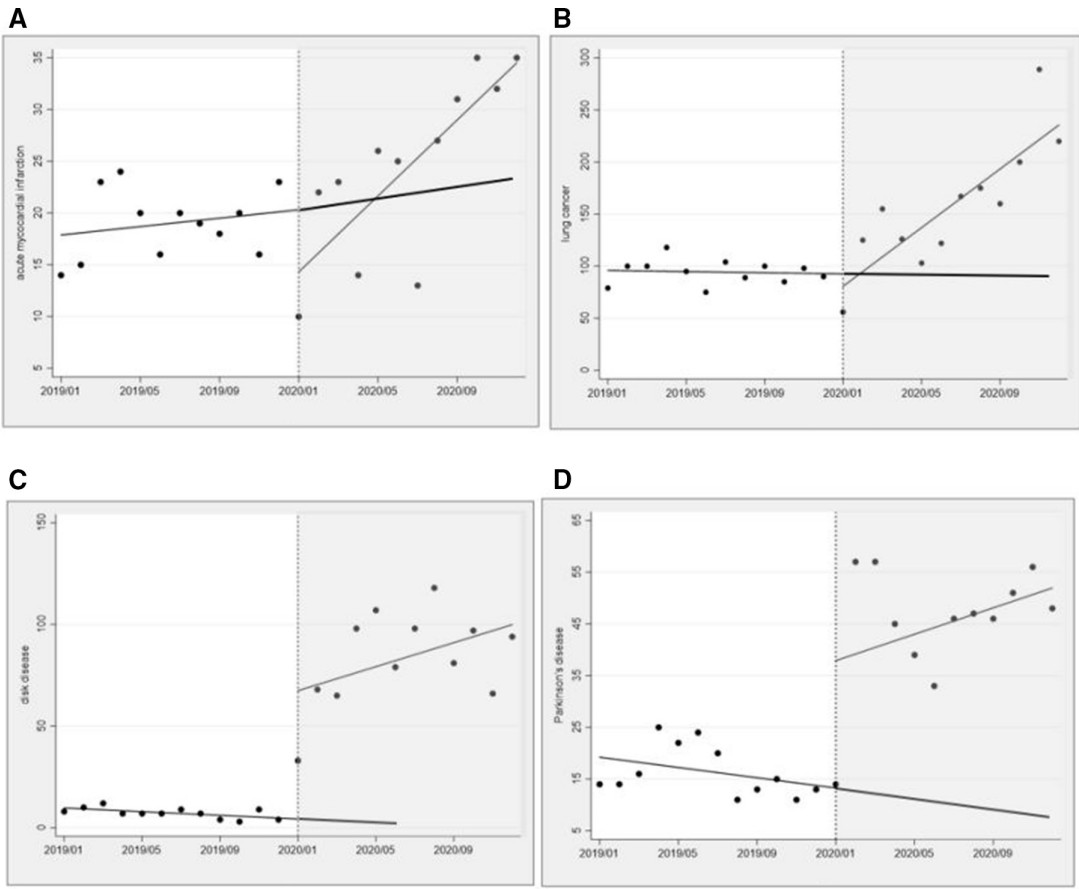

**Figure 6** Effect of COVID-19 intervention on online-consultation utilisation by disease type. Scatter plots represent monthly medical utilisation. The black line of best fit represents medical utilisation across estimated by postregression. The vertical black line is the intervention point. (A) AMI, (B) LC, (C) disk disease and (D) Parkinson's disease.

long-term one. As only 1 year of the pandemic was examined in the current study, the long-term effects here referred to change trends within 1 year. In other words, this study is a snapshot. Simultaneously, the effects of

COVID-19 on healthcare services for different diseases varied. Non-critical-disease patients showed greater reductions in face-to-face service utilisation and chose online consultations immediately after the COVID-19

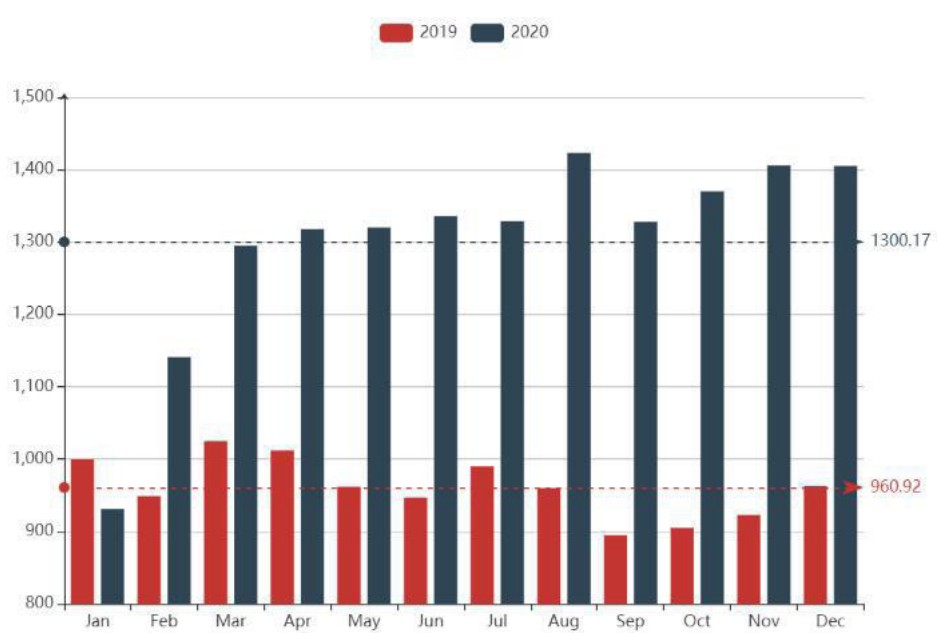

**Figure 7** Changes in online physician volume in 2020 compared with that in 2019. The dotted line is the mean.

lockdown. However, critical-disease patients chose face-to-face treatment first, and slowly moved online over time. The volume of online services provided by physicians also significantly increased as a result of COVID-19.

Our study is consistent with previous reports on hospital healthcare-service changes during the COVID-19 pandemic.[1 7] A systematic review from the UK analysed 81 studies and reported that the utilisation of healthcare in care facilities sharply decreased by about a third during the pandemic period, with greater reductions among people with less severe illnesses.[1] Additionally, according to a study from one large health system (NYU Langone Health) in the USA, telemedicine visits increased from 102.4 daily to 801.6 daily between 2 March and 14 April 2020, with a concomitant decline in in-person visits by over 80%. We also showed that online consultation underwent sustained growth owing to the impact of COVID-19, whereas face-to-face healthcare services showed a short-term decline. An important reason for the decline in face-to-face visits was that rigorous public health measures were implemented in China during the COVID-19 epidemic.[17] Some medical resources responded to the outbreak. The number of outpatient and emergency visits to 22 municipal hospitals in Beijing decreased by 36.33% in 2020. More medical personnel provided epidemic prevention and control services, such as nucleic acid tests and vaccines. Furthermore, potential risk factors in the hospital environment made people anxious and worried. A survey has shown that 20% of pregnant women fear going to a hospital for consultation, and more than 40% fear going to a hospital for check-ups.[18]

A rapid decrease in overall face-to-face discharge admissions was observed for four different diseases across 22 tertiary public hospitals in Beijing after the COVID-19 lockdown. A study performed in Germany revealed a similar reduction in traditional face-to-face healthcare services for different diseases.[19] That study found that hospital admissions for malignant cancer treatments and AMI decreased after the lockdown announcement. However, a difference was that our analysis also showed the impact of the epidemic on diseases with varied severity. We also examined the changes in online consultations for different diseases. We found that different diseases showed different online change trends. Online consultations for critical illnesses (such as AMI and LC) did not increase immediately after the outbreak but gradually increased in the long-term study. A study has shown that telemedicine can be used to reach patients with severe disease burdens.[20] Conversely, we observed that non-critical clinical situations (such as disk disease and Parkinson's disease) considerably and immediately increased, and this trend has been maintained.

Several reasons may explain the dramatic decrease in face-to-face healthcare services after the COVID-19 pandemic. First, lockdown policy and population mobility control measures were implemented in China to mitigate the spread of the pandemic. For instance, the community implemented closed management, hospital services were limited and non-emergency services had to be reserved in advance and staggered used in Beijing during the pandemic period. This situation may have reduced the availability and accessibility of face-to-face healthcare services. Second, some patients chose to postpone their treatment to prevent cross-infection in the hospital. Most healthcare resources were relocated to manage the pandemic. This setup may trigger attitudes towards delaying less urgent cases by the patient and the healthcare system. Third, some health demands may have been eliminated. For instance, many patients with self-resolving diseases chose not to avail of face-to-face treatment, such as for colds and diarrhoea.[21 22] The more important point was that some health demands may have been transferred online. Abundant internet diagnosis and treatment services were supplied by medical institutions in Beijing during the pandemic, such as online consultation, instant messaging, synchronous chat and video conferencing. Online consultation can span time and space. The physician and patient do not have to meet each other within a physical space, subsequently decreasing the requirement for face-to-face travel, appointment-related concerns and the risk of face-to-face infection. Some patients could migrate to online consultation to meet their health needs, which can explain the significantly increased volume of online consultations.

On the one hand, our results indicated that mass migration to online consultations occurred shortly after the COVID-19 pandemic. On the other hand, the migration of critically ill patients was slower than that of non-critically ill ones because the latter relied more on facility treatment. For example, the efficiency of the care pathway at every step matters very much for AMI patients.[19] Chemotherapy, radiation therapy, surgery and molecular targeted therapy are keystones in the treatment of early and locally advanced LC with good prognosis.[23] These conditions rely on face-to-face operations and inspections, and online consultations cannot provide these services, which is why the online-consultation volume for critical illnesses did not increase immediately after the outbreak. Additionally, we found that the online-consultation volume for three diseases, except for Parkinson's, continuously increased. This phenomenon can be explained by the fact that the COVID-19 pandemic accelerated the popularity of online healthcare, thereby increasing the awareness and acceptance of the public regarding online healthcare.

Our study also found that the pandemic led to a significant increase in online patients and also facilitated the online migration of physicians. The number of physicians offering services online in 2020 was much higher than that over the same period in 2019, except for January. This finding showed that physicians paid attention to providing online healthcare services.

With the COVID-19 pandemic, the supply of face-to-face healthcare services was relatively insufficient, resulting in partial healthcare-service disruption. The COVID-19 pandemic resulted in decreased supply of face-to-face

healthcare services, which was part of the healthcare-service disruption. To reduce the public-health impact, we put forward two suggestions: first, we should designate some part of healthcare resources to online provision. Moreover, we should rapidly establish a collaborative model of face-to-face and online healthcare services in public-health emergencies to meet the demand of patients. Online consultations and telemedicine can alleviate geographic and physician shortages, expand healthcare accessibility and improve the quality of healthcare provision. Therefore, online healthcare services can serve as an important supplement to traditional healthcare services, optimise health resource allocation and improve service efficiency, especially for noncritically ill patients.

However, not all diseases are suitable for online consultations. Face-to-face healthcare services are necessary and irreplaceable, such as physical examinations, diagnoses and treatment of difficult and severe diseases. Different healthcare support measures should be formulated according to the characteristics of different diseases. For critical cases, face-to-face channels ought to be opened for diagnosis and treatment to ensure that patients can receive face-to-face treatment in a timely manner. Studies indicate that 9.1% of patients experienced a delay in LC treatment, and half of AMI patients did not reach out to the hospital at all during the COVID-19 pandemic.[19 24] For critical cases, delaying or stopping treatment can have a significant negative impact on the physical and mental health of critically ill patients. A timely diagnosis and improved treatment strategy will facilitate the health of patients.[23] For mild cases, cooperation among the government, health facilities and third-party platforms should be enhanced. Online healthcare services such as telemedicine, online consultation, psychological counselling and chronic disease revisits ought to be expanded. Expanding the supply of online healthcare resources and helping patients migrate online to receive healthcare services are crucial. Meanwhile, a comparison between the quality of online consultation and face-to-face care is worth discussing. An in-depth review shows that online consultation is a useful, high-quality and safe option for the vast majority of paediatric neurology patients.[25] A study has shown that no significant difference exists between the satisfaction of online counselling and face-to-face healthcare.[26] However, studies have also shown that people still worry about the reliability of online healthcare.[18]

Finally, our work confirmed that the increase in online consultations was a persistent phenomenon during the research period. A new normal deployment of online healthcare services occurred in the pandemic and policy-enabling environment. The development of online consultation should be encouraged because of the advantages of online consultation across time and space. New technologies such as 5G technology, sensing techniques and artificial intelligence are bound to be fully exploited for online consultation. Innovating the long-term synergistic modalities of online and face-to-face healthcare services is important to promote the connection among high-quality medical resources at all levels.

This study had some limitations. First, we studied the complementary trend of online and face-to-face health service utilisation. Whether the change in service delivery caused an impact on the quality or the equivalency of care delivery requires further study. Second, to observe the changes in healthcare-service utilisation, the behaviour of patients with diseases of varied severity, that is, AMI, LC, disk disease and Parkinson's disease, was selected in this paper. Further studies are needed to determine whether this phenomenon can be extended to other diseases. The heterogeneity of the data across the hospitals was also not explored. Third, the usage of GoodDoctor cannot cover all the online-consultation behaviour. Although GoodDoctor is the largest online consultant platform in China, some other online-consultant platforms are available in the industry. We choose the GoodDoctor platform by considering that all the 22 surveyed hospitals had a certain number of doctors in the platform, which was representative to a certain extent. Finally, this study was only a snapshot capturing the first year of the pandemic response. Comprehensive research over a longer period is required.

## CONCLUSION

Face-to-face healthcare services dropped precipitously and online consultations rose sharply in 22 tertiary public hospitals in Beijing during the COVID-19 pandemic. However, face-to-face healthcare services showed a relatively fast recovery, and online consultations continued to grow. The changing trend in online-consultation volume varied by disease, which was characterised by a non-significant increase in the short term and a significant increase in the long term regarding critical cases after the outbreak, as well as a significant increase in non-critical cases in the long short term. The availability of online consultations should be encouraged during pandemics. Particular attention should be given to severely ill patients who need adequate and prompt face-to-face hospital care.

**Contributors** The research idea originated with CM, but all authors designed the study. SZ directed the data analyses. CM and SZ cowrote this manuscript. CM is responsible for the overall content and as the guarantor.

**Funding** This study has been funded by the Natural Science Foundation of Beijing Municipality (grant number: 9222003).

**Competing interests** None declared.

**Patient and public involvement** Patients and/or the public were not involved in the design, or conduct, or reporting or dissemination plans of this research.

**Patient consent for publication** Not required.

**Ethics approval** It was approved by the ethics committee of Capital Medical University (approval No. Z2020SY126).

**Provenance and peer review** Not commissioned; externally peer reviewed.

**Data availability statement** Data are available upon reasonable request. The data underlying this article were provided by [third party] by permission. Data will shared on request to the corresponding author with permission of [third party].

**ORCID iD**
Shan Zhang http://orcid.org/0000-0002-7403-3652

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
