## [Reviewer comments · BMJ Open]

ARTICLE DETAILS

TITLE (PROVISIONAL)	How has the COVID-19 pandemic affected the utilization of online consultation and face-to-face medical treatment? An interrupted time-series study in Beijing, China.
AUTHORS	Zhang, Shan; Ma, Chengyu

VERSION 1 – REVIEW

REVIEWER	Kate McBride The University of Sydney
REVIEW RETURNED	09-Aug-2022

GENERAL COMMENTS	Thank you for the opportunity to review your manuscript investigating how the COVID-19 pandemic has affected the utilisation of online consultation and offline medical treatment in Beijing, China. Overall this is a well considered paper with a high volume of data and important findings to share for the time period examined, however it would be significantly strengthened by the authors providing clearer definitions regarding some of the terms and categories used and in being structured in line with the STROBE checklist including sub-headings. Specific feedback for each section includes: ABSTRACT: 1) the use of the term 'offline' to describe care needs to be considered. It is unclear and by using such a dichotomy, it implies the default for care is online, which is not accurate. Of course, most 'offline' care as it is used in this paper, also uses online tools and electronic medical records so its a bit confusing. It also needs to be defined early on so the reader understands how the term is being applied. The authors might consider removing the use of 'offline' and replacing it with 'in-person' or 'face-to-face'. This suggestion applies throughout the manuscript.2) the 'Setting' in the Abstract needs to provide more information. From where is the data being sourced. There appears to be different sources for the online and offline activity and this needs to be described.3) the 'Participants' section as it is written needs to be moved into the results section. This should describe the eligibility criteria of the patients, how the patients or hospitals were selected, the sources including if the 22 hospitals are the same or how they are different etc4) attention needs to be given to the numerical formatting to be consistent
---

INTRODUCTION

5) a reference needs to be provided for the second sentence (lines 13 and 14)

6) it would be helpful to define 'short and long-term' pattern changes as it is not clear what the authors consider short or long

METHODS

7) As above, this section would benefit significantly from following the STROBE checklist

8) within the 'data collection' sub-heading, it would be helpful for the authors to define the following:

- selection criteria of patients e.g. was this adults only?
 - if the discharges are from acute inpatient admissions (or other care type e.g sub-acute)
 - how 'offline' healthcare is defined and as above, you may consider changing this term
 - what you consider to be 'acute' and 'non-critical' of the four diseases you examined. This needs to be defined for the reader early on.
 - how you classified or captured the diseases? e.g. was this via ICD diagnosis code or some other standard variable
 - more information about the hospitals involved e.g. were they all acute hospitals, surgical and medical, obstetrics etc it isn't clear what type of care they provided and how heterogenous the data would be
- 9) the study design section focuses on the data analysis and would be better with a different heading (in line with STROBE checklist)

RESULTS

10) the authors might consider combining some of the tables, particularly 4 and 5

11) the authors should ensure they are not duplicating information by describing in words, what is displayed in the tables e.g. Table 4

DISCUSSION

12) it would strengthen the discussions for the conclusions regarding long-term effects to be put into context regarding a) only one year of the pandemic is examined in this study and b) the pandemic is not yet over or up-to-date data included within this paper (it is a snapshot)

13) an important reason to explore regarding why there was a decrease in offline healthcare services was because the services might not have been available e.g. reduction in surgery or were diverted to the COVID-19 response. This could be further discussed.

14) a reference is needed for the statement regarding patients not choosing to get treatment face to face for colds (line 36)

15) it would strengthen the paper to further explore patient outcomes and link them to the findings of this study and the literature. For example, discuss whether the move to online services provided equivalent care, drawing on the findings of other studies. Could a decrease in offline care lead to poorer patient outcomes.

CONCLUSION

16) suggest the use of 'post-pandemic' is removed given the pandemic continues and this study is just a snap-shot in time

LIMITATIONS

17) this could be strengthened by adding the following points:

- the paper did not explore patient outcomes and whether the change in service delivery has caused an impact on their outcomes

	or the equivalency of care delivery  - it is a snap-shot in time, capturing the first year of the pandemic response. Limited conclusions can be drawn from this given what has transpired since. - heterogeneity of the data across the hospitals
--	--

REVIEWER	Chaofan Li Shandong University, School of Health Care Management
REVIEW RETURNED	24-Oct-2022

GENERAL COMMENTS	This is a very interesting paper, examine the both online and offline healthcare utilization before and after COVID-19 pandemic in Beijing. Not surprisingly, the volume of online healthcare visits increased while that of offline visits decreased largely. Some lessons could be drawn from this study as suggested in this manuscript, such as complementary relationship between online and offline healthcare. Nevertheless, the paper in its current form requires further improvements in terms of explicating the effect of COVID-19 pandemic on healthcare utilization, tidying up some of the methods, results and discussions, as well as improving English language expressions. Some suggestions for improvements are given below.  1. The literature review is a bit thin, which does not cover the latest papers on healthcare utilization after COVID-19 pandemic in China, such as “Effect of COVID-19 on hospital visits in Ningbo, China an interrupted time-series analysis”, “Effects of COVID-19 on telemedicine practice patterns in outpatient otolaryngology”, “Assessment of Pediatric Outpatient Visits for Notifiable Infectious Diseases in a University Hospital in Beijing During COVID-19”, “Impact of the COVID-19 Pandemic on Outpatient Service in Primary Healthcare Institutions An Inspiration From Yinchuan of China”, and so on. 2. Some important details of the data are not reported and should be supplemented in methods: 1) the reasons for selecting 22 municipal tertiary first-class public hospitals as study sample; 2) the characteristics of 22 tertiary public hospitals and their representativeness for the whole hospitals; 3) the reasons for including four kinds of disease into analysis; 4) the reasons for selecting GoodDoctor as data source to obtain online consultation data; 5) the introduction of GoodDoctor should cited relevant literature. 3. The authors reported COVID-19 pandemic status in Beijing in Page 6, lines 18-41. This should be placed in the introduction section. 4. I recommend the results should be reported according to the guideline of interrupted time series studies proposed by Simon L Turner: (1) Design characteristics and statistical methods used in interrupted time series studies evaluating public health interventions: protocol for a review; (2) Creating effective interrupted time series graphs: Review and recommendations. 5. “*” is seldom used to display standard error in tables. 6. Some discussion has very weak relation with the result and we cannot draw relevant conclusion, such as Page 11, lines 11-14 “This shows that physicians pay attention to providing online healthcare services during their leisure time as well, to meet the balance of medical resources between supply and demand during and after the pandemic”, and line 49 “our work confirms that the increase in online consultations is a persistent phenomenon”. 7. The manuscript should be proofread by a native English speaker for the use of English.
---

	8. Table 2 and Figure 3 report same contents and one of them should be reserved.
--	--

REVIEWER	Shasha Yuan Institute of Medical Information & Library, Chinese Academy of Medical Sciences & Peking Union
REVIEW RETURNED	27-Oct-2022

GENERAL COMMENTS	Thanks for the opportunity to review this interesting paper. The authors assessed the impact of COVID-19 pandemic on the utilization of hospital visits and online consultation during 2019-2010 in Beijing. Generally, the data is comprehensive and the results are well described. Here are some comments for further improvement.  1. It is better to clearly specify the aims of the study in the introduction part. 2. In the methods part, I suggest to specify the reasons for the four selected diseases and the basic information of the 22 public hospitals, such as how many is comprehensive hospitals and how many is infectious disease hospitals? It is unclear now. 3. Considering the sample is large enough, Mean (SD) is OK for the descriptive statistics. I think it is redundant to provide median, q1 and q3. 4. The limitation is better to be put in the discussion part.  1. It is better to clearly specify the aims of the study in the introduction part. 2. In the methods part, I suggest to specify the reasons for the four selected diseases and the basic information of the 22 public hospitals, such as how many is comprehensive hospitals and how many is infectious disease hospitals? It is unclear now. 3. Considering the sample is large enough, Mean (SD) is OK for the descriptive statistics. I think it is redundant to provide median, q1 and q3. 4. The limitation is better to be put in the discussion part.
---

REVIEWER	Zhiyuan Hou School of Public Health, National Key Laboratory of Health Technology Assessment (Ministry of Health), Collaborative Innovation Center of Social Risks Governance in Health, Fudan University
REVIEW RETURNED	04-Nov-2022

GENERAL COMMENTS	The impact of pandemic on health services needs more attentions. This is a well-design study to assess the pandemic's impact using Beijing as a case. I have some concerns to address before publication. More evidence on the impact of pandemic on health service utilization should be summarized in the introduction section. Did the 22 hospitals covered all tertiary first-class public hospitals managed by the Beijing Municipal Health Commission? Otherwise, the authors should state their sampling tech. It should state the reasons for selecting four diseases including acute myocardial infarction, lung cancer, disk disease, and Parkinson's disease. GoodDoctor is one of the online medical platforms. Its usage cannot
---

	cover all the online consultation, which should be mentioned as a limitation. The reference needs to be updated. English language should be improved throughout the manuscript.
--	--

VERSION 1 – AUTHOR RESPONSE

Reviewer #1:

1. Overall this is a well-considered paper with a high volume of data and important findings to share for the time period examined, however it would be significantly strengthened by the authors providing clearer definitions regarding some of the terms and categories used and in being structured in line with the STROBE checklist including sub-headings.

Response: Thanks for the reviewers' suggestion, the STROBE checklist is an important aid to our research. We have conducted a self-examination to improve the research and upload the STROBE checklist.

2. ABSTRACT:

(1) the use of the term 'offline' to describe care needs to be considered. It is unclear and by using such a dichotomy, it implies the default for care is online, which is not accurate. Of course, most "offline" care as it is used in this paper, also uses online tools and electronic medical records so it's a bit confusing. It also needs to be defined early on so the reader understands how the term is being applied. The authors might consider removing the use of 'offline' and replacing it with 'in-person' or 'face-to-face'. This suggestion applies throughout the manuscript.

Response: It is really true as the reviewer mentioned that the use of the term 'offline' is unclear. So we replace 'offline' with 'face-to-face' in the full text.

(2) the 'Setting' in the Abstract needs to provide more information. From where is the data being sourced. There appears to be different sources for the online and offline activity and this needs to be described.

Response: We are sorry that we may have not expressed it clearly. The data sources online and offline were supplemented in the 'Setting' section. The face-to-face healthcare service utilization data were from 22 municipal tertiary public hospitals in Beijing. The online consultations data were obtained from GoodDoctor (www.haodf.com) which is the largest and longest-running online physician-patient community in China.

(3) the 'Participants' section as it is written needs to be moved into the results section. This should describe the eligibility criteria of the patients, how the patients or hospitals were selected, the sources including if the 22 hospitals are the same or how they are different etc.

Response: Thank you for your valuable comment, we have corrected according to your comment. We have added the Participants section into the part of Methods. 22 hospitals were selected in this study, which covered all tertiary first-class public hospitals managed by the Beijing Hospital Authority. There are 13 general hospitals and 9 specialized hospitals. Specialties include obstetrics and gynecology, pediatrics, oncology, stomatology, chest, psychiatry, and traditional Chinese medicine. We collected outpatient and emergency visits, and the number of discharges from all 22 Beijing municipal hospitals, patients including adults and children.

(4) attention needs to be given to the numerical formatting to be consistent

Response: We are very sorry for our negligence of the numerical formatting mistakes. The numerical formatting has been adjusted to be consistent.

3. INTRODUCTION

(1) a reference needs to be provided for the second sentence (lines 13 and 14)

Response: Thank you very much for your reminder. The corresponding references have been added.

(2) it would be helpful to define 'short and long-term' pattern changes as it is not clear what the authors consider short or long

Response: It is really true as your suggestion. The definition of short and long-term pattern changes is supplemented. The change of short-term refers to the instantaneous change in medical service utilization at the beginning of the pandemic from the end of December 2019 to February 2020, while the long-term refers to a snapshot change within the first year of the pandemic in 2020.

4. METHODS

(1) As above, this section would benefit significantly from following the STROBE checklist

Response: Thanks for the reviewers' suggestion, and we have checked and modified this part of the paper according to the STROBE checklist.

(2) within the 'data collection' sub-heading, it would be helpful for the authors to define the following:

- selection criteria of patients e.g. were this adult only?

Response: Thanks for your reminder. 22 hospitals were selected in this study, which covered all tertiary first-class public hospitals managed by the Beijing Hospital Authority. There are 13 general hospitals and 9 specialized hospitals (obstetrics and gynecology, pediatrics, oncology, stomatology, chest, psychiatry, traditional Chinese medicine, etc.). We collected outpatient and emergency visits, and the number of discharges from all 22 Beijing municipal hospitals. Patients including adults and children, because there is a children's hospital.

- if the discharges are from acute inpatient admissions (or other care type e.g sub-acute)

Response: We are very sorry that we may have not expressed it clearly. Discharged patients refer to patients who are transferred from outpatient or emergency departments to inpatient departments, including acute, chronic and severe cases.

- how 'offline' healthcare is defined and as above, you may consider changing this term

Response: It is really true. We replace 'offline' with 'face-to-face' in the full text.

- what you consider to be 'acute' and 'non-critical' of the four diseases you examined. This needs to be defined for the reader early on.

Response: We supplement the definitions of four disease types in the methods section. Acute myocardial infarction and lung cancer were considered severe cases, while disk disease and Parkinson's disease were considered mild cases in the research.

- how you classified or captured the diseases? e.g. was this via ICD diagnosis code or some other standard variable

Response: We are sorry that we may have not expressed it clearly. We use The International Statistical Classification of Diseases and Related Health Problems, Tenth Revision (ICD-10) to identify the principal diagnoses based on the main diagnosis of the electronic medical record system of hospitals.

- more information about the hospitals involved e.g. were they all acute hospitals, surgical and medical, obstetrics etc it isn't clear what type of care they provided and how heterogenous the data would be

Response: Thank you very much for your suggestion. The information about the hospitals was involved in the Participant section. 22 hospitals were selected in this study, which covered all tertiary first-class public hospitals managed by the Beijing Hospital Authority. Including 13 general hospitals and 9 specialized hospitals (obstetrics and gynecology, pediatrics, oncology, stomatology, chest, psychiatry, traditional Chinese medicine, etc.).

(3) the study design section focuses on the data analysis and would be better with a different heading (in line with STROBE checklist)

Response: It is valuable to your suggestion. We have reassigned titles based on the STROBE checklist, including the 'Study design' and 'Statistical methods' sections.

5. RESULTS

(1) the authors might consider combining some of the tables, particularly 4 and 5

Response: Thanks for your suggestion. We combined Tables 4 and 5, and updated as Table 3.

(2) the authors should ensure they are not duplicating information by describing in words, what is displayed in the tables e.g. Table 4

Response: Thanks for your suggestion. We removed duplicating information and optimized the writing.

6. DISCUSSION

(1) it would strengthen the discussions for the conclusions regarding long-term effects to be put into context regarding a) only one year of the pandemic is examined in this study and b) the pandemic is not yet over or up-to-date data included within this paper (it is a snapshot)

Response: As the reviewer suggested that the long-term effects are poorly defined. We strengthen the discussions for long-term effects in the part of the Discussion section. As only one year of the pandemic is examined in this study, the long-term effects here refer to change trends within one year. And described it as a snapshot.

(2) an important reason to explore regarding why there was a decrease in offline healthcare services was because the services might not have been available e.g. reduction in surgery or were diverted to the COVID-19 response. This could be further discussed.

Response: Thanks for your suggestion. We further discussed the reasons for the decline in face-to-face visits. An important reason for the decline in face-to-face visits is that rigorous public health measures were implemented in China during the COVID-19 epidemic. Some medical resources responded to the outbreak. The number of outpatient and emergency visits to 22 municipal hospitals in Beijing decreased by 36.33% in 2020. More physicians have to provide epidemic prevention and control services, such as nucleic acid tests, vaccines, etc. In addition, potential risk factors in the hospital environment make people anxious and worried.

(3) a reference is needed for the statement regarding patients not choosing to get treatment face to face for colds (line 36)

Response: Thank you very much for your suggestion. It is really true. References 23 and 24 were added.

(4) it would strengthen the paper to further explore patient outcomes and link them to the findings of this study and the literature. For example, discuss whether the move to online services provided equivalent care, drawing on the findings of other studies. Could a decrease in offline care lead to poorer patient outcomes?

Response: We added relevant content in the part of the Discussion as follows.

In the meantime, the comparison between the quality of online consultation and face-to-face care is worth discussing. An in-depth review shows that online consultation is a useful, high-quality and safe option for the vast majority of pediatric neurology patients. A study showed that there is no significant difference between the satisfaction of online counselling and face-to-face healthcare. But there are also studies showing that people still worry about the reliability of online healthcare.

7. CONCLUSION

(1) suggest the use of 'post-pandemic' is removed given the pandemic continues and this study is just a snap-shot in time

Response: It is valuable to your suggestion. We deleted the words of 'post-pandemic'.

8. LIMITATIONS

(1) this could be strengthened by adding the following points:

- the paper did not explore patient outcomes and whether the change in service delivery has caused an impact on their outcomes or the equivalency of care delivery
- it is a snap-shot in time, capturing the first year of the pandemic response. Limited conclusions can be drawn from this given what has transpired since.
- heterogeneity of the data across the hospitals

Response: We have rewritten this part according to the reviewer's comments. We added some content to the limitations. First of all, we studied the complementary trend of online and face-to-face health services utilization. Whether the change in service delivery has caused an impact on the quality or the equivalency of care delivery needs further study. Secondly, to observe the changes in healthcare service utilization, the behaviour of patients with diseases of different severity, i.e. acute myocardial infarction, lung cancer, disk disease, and Parkinson's disease were selected in this paper. Further studies need to determine whether this phenomenon can be extended to other diseases. And

the heterogeneity of the data across the hospitals was not explored. Finally, it is a snapshot in time capturing the first year of the pandemic response. The comprehensive findings over a longer period require further research.

Reviewer #2:

Dr Chaofan Li, Shandong University

1. The literature review is a bit thin, which does not cover the latest papers on healthcare utilization after COVID-19 pandemic in China, such as "Effect of COVID-19 on hospital visits in Ningbo, China: an interrupted time-series analysis", "Effects of COVID-19 on telemedicine practice patterns in outpatient otolaryngology", "Assessment of Pediatric Outpatient Visits for Notifiable Infectious Diseases in a University Hospital in Beijing During COVID-19", "Impact of the COVID-19 Pandemic on Outpatient Service in Primary Healthcare Institutions: An Inspiration From Yinchuan of China", and so on.

Response: It is valuable to the reviewer's suggestion. As the reviewer suggested that it is indeed better to give some latest references for the article. We added the latest papers on healthcare utilization after the COVID-19 pandemic, like reference 2, reference 3, reference 16, reference 19, etc.

2. Some important details of the data are not reported and should be supplemented in methods: 1) the reasons for selecting 22 municipal tertiary first-class public hospitals as study sample; 2) the characteristics of 22 tertiary public hospitals and their representativeness for the whole hospitals; 3) the reasons for including four kinds of disease into analysis; 4) the reasons for selecting GoodDoctor as data source to obtain online consultation data; 5) the introduction of GoodDoctor should cite relevant literature.

Response: Thank you for your valuable comment, we have corrected according to your comment. The types of hospitals and diseases were supplemented in the methods section, as well as the relevant literature information of GoodDoctor. 22 hospitals were selected in this study, which covered all tertiary first-class public hospitals managed by the Beijing Hospital Authority. There are 13 general hospitals and 9 specialized hospitals (obstetrics and gynecology, pediatrics, oncology, stomatology, chest, psychiatry, traditional Chinese medicine, etc.).

We stratified models by disease type restricted to (1) acute myocardial infarction, (2) lung cancer, (3) disk disease, and (4) Parkinson's disease. These are all the diseases in 20 monitoring diseases of Beijing Hospital Authority with clear diagnostic criteria. These four diseases have a certain number of patients both online and offline. Acute myocardial infarction and lung cancer were considered severe cases, while disk disease and Parkinson's disease were considered mild cases in the research.

The GoodDoctor is the largest and longest-running online physician-patient communication platform in China. It was founded in 2006. As of October 2021, it has collected information on 860,000 physicians from 9,780 regular hospitals in China, of which more than 73% are physicians in tertiary hospitals. The 22 hospitals are all included in the GoodDoctor platform, covering more than 80% of outpatient doctors of these hospitals. Reference 2 and reference 3 were supplemented.

3. The authors reported COVID-19 pandemic status in Beijing in Page 6, lines 18-41. This should be placed in the introduction section.

Response: Thanks for your suggestion. The COVID-19 pandemic status in Beijing has been moved to the part of Introduction.

4. I recommend the results should be reported according to the guideline of interrupted time series studies proposed by Simon L Turner: (1) Design characteristics and statistical methods used in interrupted time series studies evaluating public health interventions: protocol for a review; (2) Creating effective interrupted time series graphs: Review and recommendations.

Response: Thanks for your suggestion, we have improved the depiction of the time series graph according to interrupted time series studies proposed by Simon L Turner. Each interruption of the time series is shown with a vertical line and by a light shading of a period.

5. "*" is seldom used to display standard error in tables.

Response: We are sorry that we may not display it correctly. We check the paper carefully and modified the mistakes.

6. Some discussion has very weak relation with the result and we cannot draw relevant conclusion, such as Page 11, lines 11-14 "This shows that physicians pay attention to providing online healthcare services during their leisure time as well, to meet the balance of medical resources between supply and demand during and after the pandemic", and line 49 "our work confirms that the increase in online consultations is a persistent phenomenon".

Response: It is very true about your suggestion. We have removed some of the less relevant statements. And we are sorry that we may have not expressed "our work confirms that the increase in online consultations is a persistent phenomenon". We modified it to "Our work confirms that the increase in online consultations is a persistent phenomenon in the research period". It refers to a snapshot change within the first year of the pandemic.

7. The manuscript should be proofread by a native English speaker for the use of English.

Response: Thank you very much for your advice. Our manuscript has been polished by a native speaker after careful revision.

8. Table 2 and Figure 3 report same contents and one of them should be reserved.

Response: We deleted Table 2 and integrated the valid information into Figure 3.

Reviewer #3:

1. It is better to clearly specify the aims of the study in the introduction part.

Response: It is very true about your suggestion. We supplemented the aims of the study in the part of the Introduction. This study was designed to compare and analyze the changing trend of online consultation and face-to-face healthcare volume during the pandemic, as well as the change in different disease types. The research illustrates the speed, scale, and reception of online consultations utilization by patients and physicians, and the complementary trends in online and face-to-face health services. The result may help to gain insight and inform the planning of future healthcare resources during the outbreak of infectious diseases.

2. In the methods part, I suggest to specify the reasons for the four selected diseases and the basic information of the 22 public hospitals, such as how many is comprehensive hospitals and how many is infectious disease hospitals? It is unclear now.

Response: Thank you for your valuable comment, we have corrected it according to your comment. The Participants section was moved into the part of Methods. 22 hospitals were selected in this study, which covered all tertiary first-class public hospitals managed by the Beijing Hospital Authority. There are 13 general hospitals and 9 specialized hospitals. Specialties include obstetrics and gynecology, pediatrics, oncology, stomatology, chest, psychiatry, and traditional Chinese medicine.

3. Considering the sample is large enough, Mean (SD) is OK for the descriptive statistics. I think it is redundant to provide median, q1 and q3.

Response: Thank you very much for your suggestion. Median, Q1 and Q3 were deleted from the manuscript.

4. The limitation is better to be put in the discussion part.

Response: Thanks for your suggestion. The limitation was moved to the discussion part and added some content.

Reviewer #4:

1. More evidence on the impact of pandemic on health service utilization should be summarized in the introduction section.

Response: More evidence on the impact of the pandemic on health service utilization has been summarized in the introduction section, like reference 2, reference 3, reference 4, reference 5, reference 10.

2. Did the 22 hospitals covered all tertiary first-class public hospitals managed by the Beijing Municipal Health Commission? Otherwise, the authors should state their sampling tech.

Response: Thank you very much for your suggestion. The information about the hospitals was involved in the Participants section. 22 hospitals were selected in this study, which covered all tertiary first-class public hospitals managed by the Beijing Hospital Authority. There are 13 general hospitals and 9 specialized hospitals (obstetrics and gynecology, pediatrics, oncology, stomatology, chest, psychiatry, traditional Chinese medicine, etc.).

3. It should state the reasons for selecting four diseases including acute myocardial infarction, lung cancer, disk disease, and Parkinson's disease.

Response: Thank you for your valuable comment, we have corrected it according to your comment. We stratified models by disease type restricted to (1) acute myocardial infarction, (2) lung cancer, (3) disk disease, and (4) Parkinson's disease. These are the diseases in 20 monitoring diseases of the Beijing Municipal Health Commission with clear diagnostic criteria. These four diseases all have a certain number of patients both online and offline. Acute myocardial infarction and lung cancer were considered severe cases, while disk disease and Parkinson's disease were considered mild cases in the research.

4. GoodDoctor is one of the online medical platforms. Its usage cannot cover all the online consultation, which should be mentioned as a limitation.

Response: Thank you very much for your suggestion. It has been mentioned as a limitation. The usage of GoodDoctor cannot cover all the online consultation behaviour. Although GoodDoctor is the largest online consultant platform in China, there are some other online consultant platforms in the industry. We choose the GoodDoctor platform by considering the 22 survey hospitals all have a certain number of doctors in the platform, which is representative to a certain extent. Finally, this study is only a snapshot in time capturing the first year of the pandemic response. The comprehensive findings over a longer period require further research.

5. The reference needs to be updated.

Response: As the reviewer suggested that it is indeed better to give some latest references for the article. We added the latest papers on healthcare utilization after the COVID-19 pandemic, like reference 2, reference 3, reference 4, reference 5, reference 10, reference 12, reference 16, reference 17, reference 19, reference 23, reference 24, reference 25, etc.

6. English language should be improved throughout the manuscript.

Response: Thank you very much for your advice. Our manuscript has been polished by a native speaker after careful revision.

VERSION 2 – REVIEW

REVIEWER	Shasha Yuan Institute of Medical Information & Library, Chinese Academy of Medical Sciences & Peking Union Medical College
REVIEW RETURNED	15-Dec-2022

GENERAL COMMENTS	thanks for the response and I have no more comments.
--

REVIEWER	Zhiyuan Hou School of Public Health, National Key Laboratory of Health Technology Assessment (Ministry of Health), Collaborative Innovation Center of Social Risks Governance in Health, Fudan University
REVIEW RETURNED	09-Dec-2022

GENERAL COMMENTS	all my comments have been well addressed.
---